# A molecular pyroelectric enabling broadband photo-pyroelectric effect towards self-driven wide spectral photodetection

Xi Zeng [1,2], Yi Liu[1,2], Wen Weng [1,2], Lina Hua[1], Liwei Tang[1], Wuqian Guo[1,2], Yaoyao Chen[1], Tian Yang[1], Haojie Xu[1,2], Junhua Luo [1,2,3] ✉ & Zhihua Sun [1,2,3] ✉

Broadband spectral photoresponse has shown bright prospects for various optoelectronic devices, while fulfilling high photoactivity beyond the material bandgap is a great challenge. Here, we present a molecular pyroelectric, *N*-isopropylbenzylaminium trifluoroacetate (*N*-IBATFA), of which the broadband photo-pyroelectric effects allow for self-driven wide spectral photodetection. As a simple organic binary salt, *N*-IBATFA possesses a large polarization (~9.5 µC cm$^{-2}$), high pyroelectric coefficient (~6.9 µC cm$^{-2}$ K$^{-1}$) and figures-of-merits ($F_V = 187.9 \times 10^{-2}$ cm$^2$ µC$^{-1}$; $F_D = 881.5 \times 10^{-5}$ Pa$^{-0.5}$) comparable to the state-of-art pyroelectric materials. Particularly, such intriguing attributes endow broadband photo-pyroelectric effect, namely, transient currents covering ultraviolet (UV, 266 nm) to near-infrared (NIR, 1950 nm) spectral regime, which breaks the restriction of its optical absorption and thus allows wide UV-NIR spectral photodetection. Our finding highlights the potential of molecular system as high-performance candidates toward self-powered wide spectral photodetection.

Broadband photodetection from the ultraviolet (UV) to visible (vis) and near-infrared (NIR) spectral region has recently drawn booming interest for higher resolution in hyperspectral imaging, night recognition, and machine vision[1–4]. At present, a general approach to achieve broadband response is to implement materials with a wide spectral active range as light collectors[5,6]. Nonetheless, owing to light absorption limitation, single-phase photosensitive semiconductors are used to detect specific sub-bands region in the UV-vis-NIR range, e.g., GaN (250–400 nm)[7], Si (350–700 nm)[8], and InGaAs (900–1700 nm)[9]. Although different types of materials, such as perovskites, polymers, 2D materials, and heterojunctions, have shown potential in broad spectral responses, their ambient instability, preparation complexity, or high expense still hinder device overall performances[1,10–12]. To alleviate these concerns, it is critical and urgent to exploit fresh candidate alternatives or supplements with photosensitive features covering UV-vis-NIR spectral regime.

Pyroelectric effect generally exists in polar dielectric materials, which causes internal polarization variation through external temperature fluctuations to achieve pyro-to-electron conversion[13,14], and can thus be utilized in infrared detection, energy harvesting, and thermal imaging[15–17]. This physical principle is closely relevant to spontaneous polarization ($P_s$) of pyroelectric, i.e., the built-in electrostatic field would promote the separation of charge carriers[18,19]. In the past decades, inorganic oxides of PbTiO$_3$, Pb(Zr,Ti)O$_3$, and PMN-PT constitute the mainstay of pyroelectric systems owing to their excellent pyroelectric performances[20–22]. However, the majority of these

[1]State Key Laboratory of Structural Chemistry, Fujian Institute of Research on the Structure of Matter, Chinese Academy of Sciences, Fuzhou, Fujian 350002, China. [2]University of Chinese Academy of Sciences, Beijing 100039, China. [3]Fujian Science & Technology Innovation Laboratory for Optoelectronic Information of China, Fuzhou, Fujian 350108, China. ✉e-mail: jhluo@fjirsm.ac.cn; sunzhihua@fjirsm.ac.cn

materials contain environmentally harmful metals and suffer from adverse issues, such as high-temperature synthesis and integration challenge. Consequently, the urgent demand for green environment and sustainable development motivates further targeted exploration of non-toxic, low-cost pyroelectric alternatives. Among them, organic molecular pyroelectrics deserve great interest in terms of their unique merits of light weight, structural flexibility, biocompatibility, and ease of processing[23–25]. Currently, pyroelectric properties of some organic molecules almost catch up with those of typical inorganic counterparts. For instance, triglycine sulfate (TGS) has been well-used as a commercial pyroelectric material since the discovery of its pyroelectricity in 1950s[26,27]. In this context, if the performances improved, it is possible to envision that metal-free molecular pyroelectrics serve as a promising alternative class.

In principle, such $P_s$-directed pyroelectric effect could also be excited through light irradiation, termed photo-pyroelectric effect. This is a thermal process almost independent on the optical absorption of pyroelectric materials, thus allowing the wide spectral photoresponse beyond their energy bandgap[28,29]. For example, Ag$_2$Se/$p$-Si heterojunction enables photoactivity in the spectral region of 405-1064 nm via pyroelectric-photovoltaic coupling effect[30]. Currently, most of these devices were fabricated with complex $p$-$n$ junctions or Schottky junctions, while the single-phase pyroelectric candidates remain very scarce for achieving broadband response covering UV-NIR spectral region[31,32]. Furthermore, the photo-pyroelectric detector exhibits a substantially quicker response in comparison with the thermal-based detector, which might attribute to the direct correlation between photo-pyroelectric signal and temperature change, and its physical process does not necessitate the establishment of thermal balance[33]. In this study, we initially select a suitable polar component of $N$-isopropylbenzylamine ($N$-IBA), which is conducive to inducing the occurrence of ferroelectric phase transition and symmetry breaking[34]. The order-disordering feature of CF$_3$COO$^-$ cation also facilitates the phase transition. Meanwhile, the F substitution has been proven as an effective strategy to achieve the relatively high Curie temperature ($T_c$)[35,36].

Here, we present a molecular pyroelectric, $N$-isopropylbenzylaminium trifluoroacetate ($N$-IBATFA), of which the photo-pyroelectric effect drives broadband spectral photoresponse from 266 to 1950 nm. It has a large room-temperature polarization of ~9.5 μC cm$^{-2}$, high pyroelectric coefficient ($P_e$ = 6.9 μC cm$^{-2}$ K$^{-1}$) and figures-of-merit ($F_V$ = 187.9 × 10$^{-2}$ cm$^2$ μC$^{-1}$; $F_D$ = 881.5 × 10$^{-5}$ Pa$^{-0.5}$) around the $T_c$ of 360 K. Strikingly, due to light-induced pyroelectric effect, $N$-IBATFA allows broad light sensitivity in the UV-NIR spectral regime with high photoresponsivity ($R$) and detectivity ($D$*) at zero bias (i.e., self-driven mode). This work hints the potential of molecular systems as promising candidates toward self-powered photodetection.

## Results

### Phase transition behaviors

Colorless rod-like crystals of $N$-IBATFA were easily obtained by slowly evaporating its aqueous solution (Supplementary Fig. 1), and the phase purity was solidly verified by powder X-ray diffraction that agrees with the simulated pattern (Supplementary Fig. 2). Preliminary differential scanning calorimetry (DSC) and specific heat ($C_p$) measurements confirmed its phase transition (Fig. 1a), as shown by the endothermic/exothermic peaks at 360 and 324 K ($T_c$). The large thermal hysteresis of ~36 K suggests the first-order characteristic of phase transition[24,37]. The $C_p$-$T$ curve also displays a sharp peak at 360 K upon heating, being consistent with the DSC result. The strong temperature-dependent dielectric activities at different frequencies further confirm the phase transition of $N$-IBATFA around $T_c$ (Fig. 1b). In addition, Fig. 1c manifests the variable-temperature behavior of second harmonic generation (SHG) signal, revealing the symmetry breaking occurring in phase transition. SHG-active peak below $T_c$ suggests its non-centrosymmetric

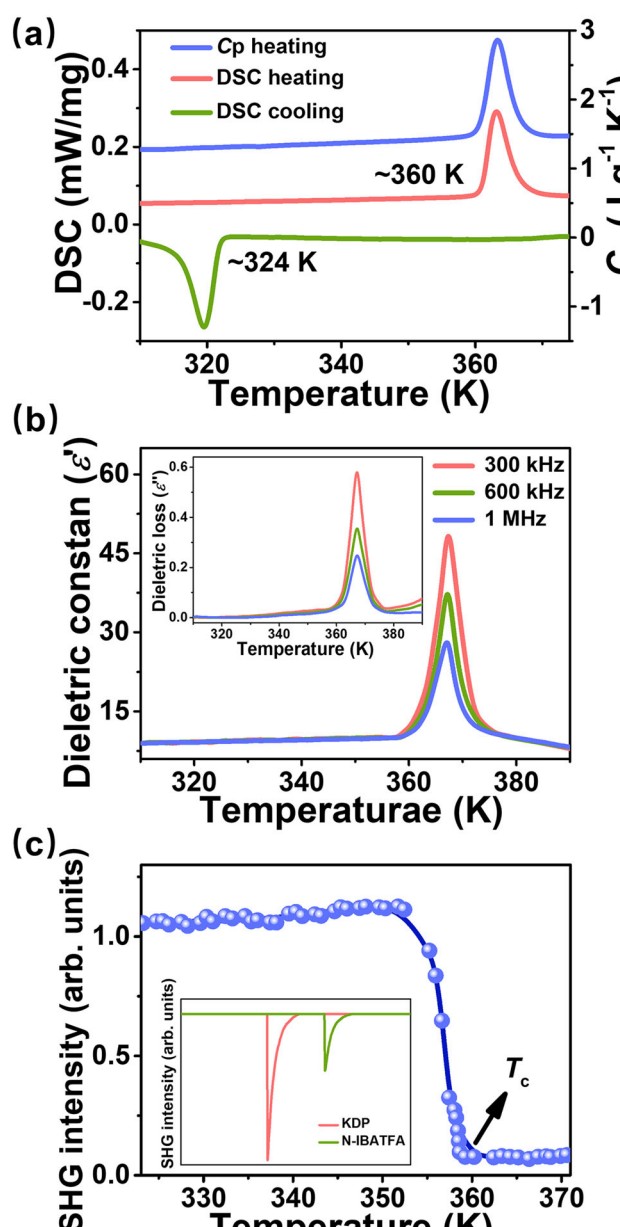

**Fig. 1 | Phase transition behaviors of $N$-IBATFA. a** DSC and $C_p$-$T$ curves of $N$-IBATFA. **b** dielectric constant ($\varepsilon'$) and the corresponding dielectric loss ($\varepsilon''$, inset) at different frequencies upon heating. **c** Variable-dependent SHG response of $N$-IBATFA upon heating. Inset shows comparison of SHG intensities of $N$-IBATFA and KH$_2$PO$_4$ (KDP) at room temperature.

structure at low-temperature phase (LTP); with the temperature rising above $T_c$, SHG signal becomes zero, corresponding to the centrosymmetric structure at high-temperature phase (HTP)[38,39].

### Variable-temperature crystal structure analyses

To deeply investigate the structural phase transition, crystal structures of $N$-IBATFA at 293 K (LTP) and 365 K (HTP) were determined by single-crystal X-ray diffraction, respectively (Supplementary Table 1). The basic structural unit is an H-bonding dimer containing the organic $N$-isopropylbenzylammonium cation and trifluoroacetate anion (Fig. 2a and Supplementary Fig. 5), linked by strong intermolecular N-H···O hydrogen bonds (Supplementary Table 2). At LTP, $N$-IBATFA belongs to the monoclinic crystal system with a polar space group of $P2_1$, and both components are ordered with small thermal ellipsoids

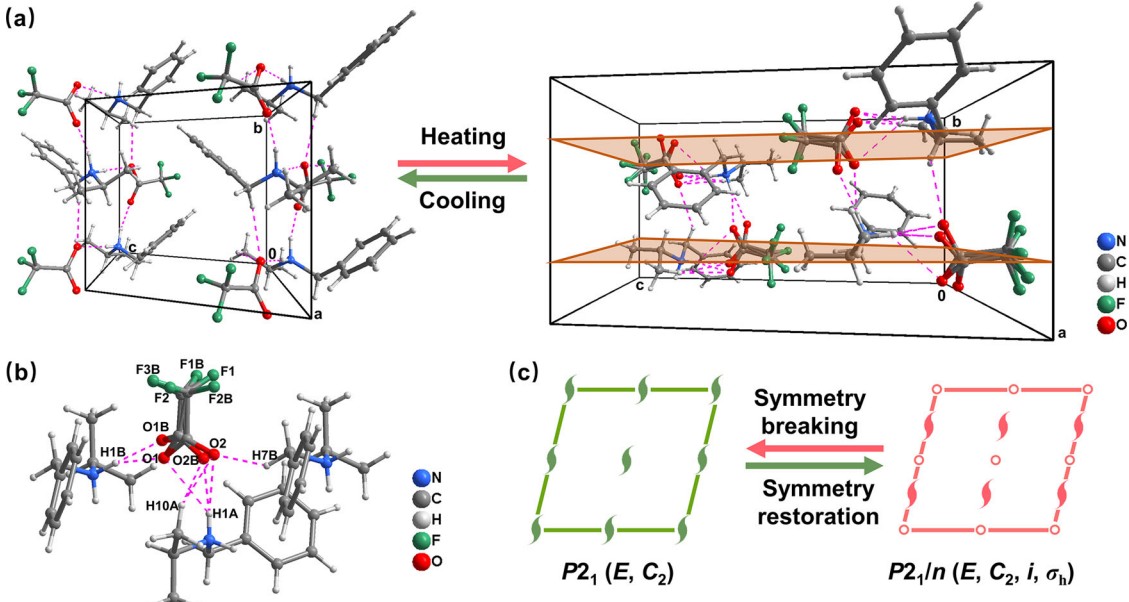

**Fig. 2 | Crystal structures of *N*-IBATFA. a** The reversible phase transition between LTP (left) and HTP structures (right) (H-bonds and glide planes are represented by magenta the dotted lines and orange planes, respectively). **b** Hydrogen bonds of *N*-IBATFA at HTP. **c** Symmetry breaking with an Aizu notation of 2/*mF*2.

(Supplementary Figs. 6–7, Supplementary Tables 3–4). As temperature increases above $T_c$, *N*-IBATFA still crystallizes in the monoclinic crystal system but the space group turns into the centrosymmetric one of $P2_1/n$, revealing the occurrence of phase transition. As shown in Fig. 2a, the HTP structure has (orange) glide plane perpendicular to (010) with glide component (0.5, 0, 0.5) in addition to two-fold axis and inversion center. A noticeable variation is that trifluoroacetate anions become disordered (Fig. 2b) with relatively increased thermal ellipsoids (Supplementary Fig. 7). Although no distinguishable alterations are found in the cationic configuration, the thermal ellipsoids show an enlarging trend compared to those at LTP. Therefore, the disordering of dynamic anions leads to the breaking of pristine symmetry; the number of symmetric operations increases from 2 ($P2_1$, LTP) to 4 ($P2_1/n$, HTP). From the perspective of symmetry breaking (Fig. 2c), the structural phase transition of *N*-IBATFA obeys the ferroelectric transition with an Aizu notation of 2/*mF*2[40,41]. The electric polarization *versus* electric field (*P-E*) hysteresis loop of *N*-IBATFA crystal was measured along its polar axis at LTP. However, the electric switching of polarization could not be achieved under a high electric field up to ~120 kV/cm (Supplementary Fig. 8). Further studies on the domain structures are still needed in the future. Consequently, the current evidence is insufficient to conclusively confirm its ferroelectricity, while the ferroelectric phase transition indicates that *N*-IBATFA is expected to be a potential metal-free molecular pyroelectric material.

**Pyroelectric effect**

To confirm pyroelectric effect of *N*-IBATFA, pyroelectric experiments were carried out on single crystals along the polar axis direction at zero bias upon heating. As illustrated in Fig. 3a, a sharp pyroelectric current is generated with the temperature approaching $T_c$, which has the compensation to charge displacement. By integrating pyroelectric current with respect to time, $P_s$ value obtained is ~9.5 µC cm$^{-2}$ at LTP, while it vanishes above $T_c$ (Fig. 3a), verifying the occurrence of the polar-to-nonpolar phase transition. Besides, the sharply decreasing of $P_s$ affords a large $P_e$ value of ~6.9 µC cm$^{-2}$ K$^{-1}$ around $T_c$. This value exceeds several typical pyroelectric materials, such as TGS (-0.055 µC cm$^{-2}$ K$^{-1}$), LiTaO$_3$ (-0.98 µC cm$^{-2}$ K$^{-1}$), and PMN-PT (-0.3 µC cm$^{-2}$ K$^{-1}$) (Supplementary Table 5)[22,42,43]. Notably, as two essential figures-of-

merit (FOMs) to evaluate pyroelectric devices, voltage responsivity $F_V = P_e/(\varepsilon_0\varepsilon_r C_v)$ and detection capability $F_D = P_e/[C_v(\varepsilon_0\varepsilon_r \tan\delta)^{0.5}]$, where $P_e$, $\varepsilon_0$, $\varepsilon_r$, $\tan\delta$, and $C_v$ are the pyroelectric coefficient, vacuum permittivity, dielectric constant, dielectric loss, and volume-specific heat, describe the voltage output efficiency and signal-to-noise ratio limit, respectively[44]. As depicted in Fig. 3b, the temperature-dependent $F_V$ and $F_D$ exhibit a consistent behavior with $P_e$; the peak values of ~187.9 × 10$^{-2}$ cm$^2$ µC$^{-1}$ and ~881.5 × 10$^{-5}$ Pa$^{-0.5}$ are observed in the vicinity of $T_c$, far beyond some conventional pyroelectrics (e.g., TGS, DTGS, and PMN-PT) (Supplementary Table 5)[22,41,45,46]. Although $F_V$ (7.0 × 10$^{-2}$ cm$^2$ µC$^{-1}$) and $F_D$ (6.1 × 10$^{-5}$ Pa$^{-0.5}$) of *N*-IBATFA at room temperature are inferior to peak values near $T_c$, these two FOMs are still comparable to other molecular pyroelectrics even ferroelectrics owing to its small dielectric constant. Overall, the obtained figure-of-merits, including $F_V$ and $F_D$, make *N*-IBATFA a potential material for pyroelectric application as the molecular alternative, and its non-toxic, light-weight and environmentally friendly advantages render it a valuable reference for the development of flexible and biocompatible devices.

**Broadband photo-pyroelectric effect**

To further illustrate the sensibility to small fluctuations in temperature, we performed pyroelectric experiments at room temperature. Here, a 405 nm laser with 160 mW cm$^{-2}$ was chosen as the heat source to irradiate the crystal of *N*-IBATFA, and the induced pyroelectric current varies simultaneously with a temperature change of 0–4.5 K generated by alternatively switching light (Fig. 3c, d). The cyclic changes in temperature and the corresponding differential temperature $dT/dt$ are shown in Supplementary Fig. 9. As shown in Fig. 3e, when the temperature is kept constant ($dT/dt = 0$), electric dipoles within pyroelectric maintain stability and there is no electricity in the external circuit. While material temperature goes up during laser irradiation ($dT/dt > 0$), breaks the original equilibrium of electric dipoles, with the result that the free charges on the material surface decrease due to reduced $P_s$, thus generating a pyroelectric potential across the material[16]. Then the electric current will emerge in the circuit, showing a sharp positive transient current spike ($I_{pyro}$) (Fig. 3d). As the temperature gradually stabilizes (illumination remains), pyroelectric current rapidly reaches a stable platform and disappears. On the contrary, when the temperature

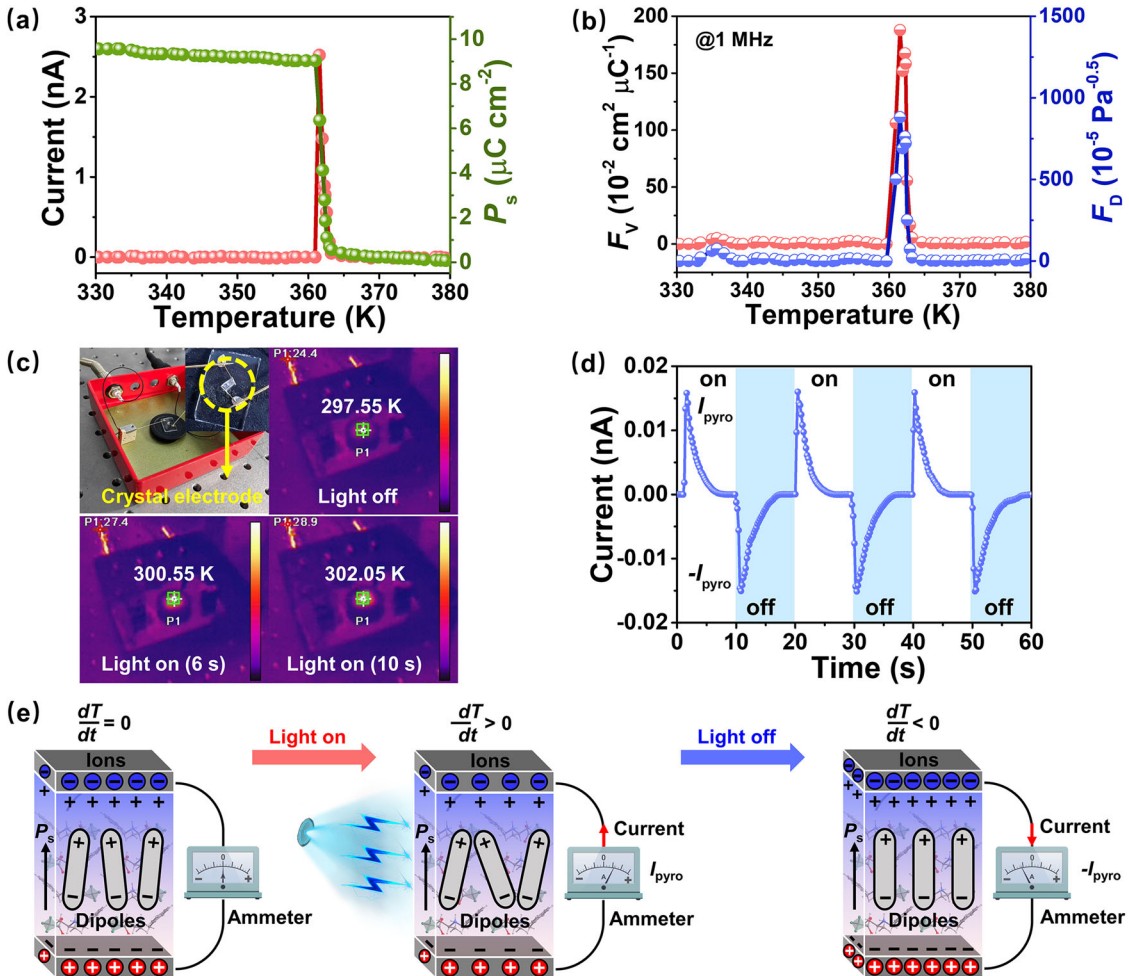

**Fig. 3 | Pyroelectric and photo-pyroelectric properties of *N*-IBATFA.**
**a** Temperature dependence of pyroelectric current and $P_s$ determined by integrating pyroelectric current over time. **b** The simulated performances of two $F_V$ and $F_D$ FOMs. **c** Thermal imaging photos of crystal surface in the dark and irradiated by 405 nm laser. **d** Pyro-generated current responses under a periodically switched 405 nm laser. **e** Schematic diagram of the principal mechanism of photo-pyroelectric effect.

suddenly drops due to the light being turned off, an equivalent negative transient current spike ($-I_{pyro}$) is induced. With material temperature returning to ambient temperature, the pyroelectric polarization field disappears, resulting in a dark current.

The working principle of *N*-IBATFA for self-driven photodetection via light-to-pyro-to-electron conversion is the result of a polarization-directed mechanism beyond the intrinsic energy bandgap (Supplementary Fig. 10). In other words, the high-temperature gradient induced by strong light would enhance photo-pyroelectric effect, which can greatly broaden the spectral response range of pyroelectric devices in the UV-vis-NIR region. Based on this principle, self-powered instantaneous response characteristics of *N*-IBATFA were systematically investigated at $V_{bias} = 0$ by using 266, 405, 520, 637, 785, 940, 1550, and 1950 nm lasers with different power densities as the excitation source. As depicted in Fig. 4a and Supplementary Fig. 11a, the current peak increases significantly with the increasing power density of 405 nm laser, which is owing to the larger temperature fluctuations and production of more electrons/holes under higher light intensities. Similar trends are observed for other illumination wavelengths (266, 520, 637, 785, 940, 1550, and 1950 nm), as shown in Supplementary Fig. 12. Furthermore, the dynamic response behavior, illustrated in Fig. 4b and Supplementary Fig. 11b, reveal a wavelength dependency across the entire range from 266 to 1950 nm, indicating distinct photoactivity. Additional elaboration on the potential relationship

between $I_{pyro}$ and power density or laser wavelength is provided in Supplementary Fig. 13. The highest pyroelectric current response observed at 266 nm is possibly due to the coupling with photovoltaic effect, since the absorbed UV light might excite the generation of electron-hole pairs[47,48]. Moreover, to further verify the photo-pyroelectric behaviors, we conducted a comparative analysis of the light-induced pyroelectric responses between TGS and *N*-IBATFA. The qualitative results indicate that *N*-IBATFA exhibits photo-pyroelectric response comparable to that of TGS (Supplementary Fig. 14). This unique broadband photoresponse of *N*-IBATFA, covering a wide spectral range from UV to NIR (266–1950 nm), exceeds that of several reported detectors, which facilitates its potential application in self-powered spectral detection (Supplementary Table 6).

Two crucial parameters of photoresponsivity ($R$) and detectivity ($D^*$) are calculated based on the crystal device of *N*-IBATFA to further evaluate self-powered device performance (Fig. 4c). $R$ is defined as the ratio of photo-pyroelectric current to incident light intensity, i.e., $R = I_{pyro}/P_*S$, while $D^*$ denotes the ability to detect weak light signals that can be defined using the formula $D^* = R/(2eI_{dark}/S)^{0.5}$, where $I_{pyro}$, $P$, $S$, $I_{dark}$, $e$ are the photo-pyroelectric current, incident light intensity, effective plane area of device, dark current, and charge of an electron, respectively[49]. Figure 4d shows transient $R$ and $D^*$ of *N*-IBATFA at weak light power density of 405 nm laser. As power density decreases, $R$ values display an increasing trend, and $D^*$ values show a similar

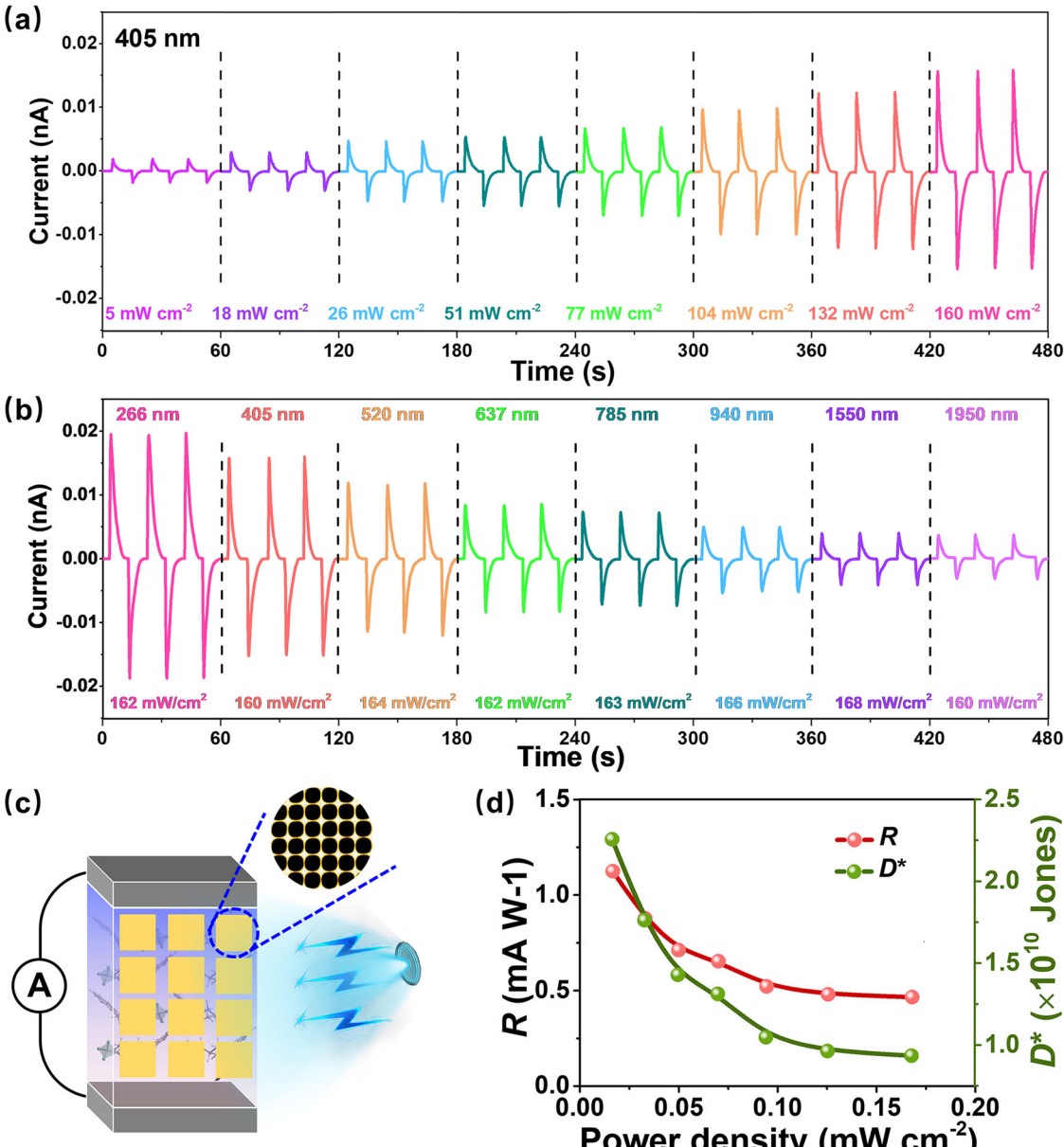

**Fig. 4 | Temporal photoresponse behaviors of *N*-IBATFA. a** Variations of current intensity as a function of power density at 405 nm. **b** Photo-pyroelectric currents under different laser wavelengths of 266–1950 nm with similar light densities.

**c** Schematic diagram of crystal device fabricated by Au electrode sputtering method. **d** Photoresponsivity and detectivity under weak light power density of 405 nm laser.

variation, with their maximum values estimated to be -1.1 mA W$^{-1}$ and 2.3 × 10$^{10}$ Jones, respectively (Supplementary Table 6), verifying that light-induced pyroelectric effect can effectively enhance photoresponsivity and detectivity. In addition, the photostability and long-term environmental stability of *N*-IBATFA were investigated under 405 nm laser irradiation with 160 mW cm$^{-2}$, showing stable output signals without any significant degradation (Supplementary Figs. 15–16). These results indicate that *N*-IBATFA should be a promising candidate for self-driven broadband detection.

## Discussion

In this study, we demonstrate a molecular pyroelectric with excellent pyroelectricity, i.e., large $P_s$ (-9.5 μC cm$^{-2}$), high $P_e$ (6.9 μC cm$^{-2}$ K$^{-1}$), and FOMs ($F_V = 187.9 × 10^{-2}$ cm$^2$ μC$^{-1}$; $F_D = 881.5 × 10^{-5}$ Pa$^{-0.5}$) compared to the state-of-art pyroelectric materials. As a result, it features photoexcited pyroelectric effect in an ultra-wide spectral range of UV-vis-

NIR (266–1950 nm), which is far beyond the optical bandgap limit. Furthermore, the as-prepared detector based on a crystal device also exhibits high photoresponsivity and detectivity at zero bias. Our work provides a promising approach to extend broadband photoresponses through photo-pyroelectric effect in the organic molecular pyroelectrics, which will lead to the further development and design of molecule-based self-driven broadband photodetectors.

## Methods

### Synthesis and crystal growth

All chemical reagents were purchased from commercial sources and used without any further purification. They include N-benzylisopropylamine (97.0%, Aladdin), trifluoroacetic acid (>99.5%, Aladdin), methanol (AR, Sinopharm Chemical Reagent Co., Ltd.), and distilled water. Colorless large-size crystals of *N*-IBATFA were easily synthesized by slowly evaporating a mixed solution (containing equal

volumes of methanol and distilled water) with the equimolar N-isopropylbenzylamine and trifluoroacetic acid.

## Single-crystal structure determination

Single-crystal X-ray diffraction (SCXRD) data at different temperatures were collected on an Agilent Technologies Supernova Dual diffractometer with Mo $K\alpha$ radiation ($\lambda = 0.71073$ Å). Data collection, cell refinement, and data reduction were processed with Rigaku Crystal-Clear software. The structures were solved by direct methods and refined by the full-matrix least-squares refinements on $F^2$ using Olex2 and SHELXTL software package. All non-hydrogen atoms were refined anisotropically, and all hydrogen atoms were generated by the geometrical method and refined by using a riding model with $Uiso = 1.2$ $Ueq$ (C). Crystallographic data and structure refinement of N-IBATFA at different phases are listed in Supplementary Tables 1–4.

## Physical characterization

Powder X-ray diffraction (PXRD) patterns were collected in the $2\theta$ range of 5–50° on the Japan Rigaku MiniFlex 600 diffractometer using a Cu $K\alpha$ X-ray source ($\lambda = 1.54178$ Å). Variable-temperature PXRD patterns were performed at 298, 368, 378, and 389 K on a Japan Rigaku Ultima IV diffractometer to confirm the phase. Ultraviolet-visible (UV-vis) absorption spectra were recorded by the PE Lambda 950 spectrophotometer using a $BaSO_4$ plate as a reference (100% reflectance). The bandgap energy was estimated by using the Kubelka-Munk function[50]. Thermogravimetric (TG) analysis was conducted on the Netzsch STA449C thermal analyzer from ambient temperature to 1200 K at a heating rate of 15 K·min$^{-1}$ under a nitrogen atmosphere. Differential scanning calorimetry (DSC) and $C_p$ analyses were performed on the NETZSCH DSC 3500 Sirius instrument with the heating/cooling rates of 10 K·min$^{-1}$ under the $N_2$ atmosphere. The dielectric analyses were performed on the TongHui TH2828 impedance analyzer.

## Second harmonic generation (SHG) measurement

SHG effect was studied by the Kurtz-Perry powder test using a Fluorescence Spectrometer (FLS 920, Edinburgh Instruments) equipped with the variable-temperature system, using the unexpanded laser beam with low divergence (pumped by an Nd-doped YAG laser with 1064 nm, 5 ns pulse duration, 10 Hz repetition rate). SHG intensity of N-IBATFA was measured through a semi-quantitative comparison with that of standard $KH_2PO_4$ (KDP).

## Electrode fabrication

Along the polar axis direction of N-IBATFA single-crystal with the size of $2.5 \times 1 \times 1$ mm$^3$, the two sides were both deposited with conductive silver adhesive and drawn out with copper wires.

## Device configuration

Based on the above electrode, the crystal surface was cleaned by a nitrogen flow and covered with copper mesh. Subsequently, the Au electrode device was deposited on the blank area of copper mesh with the thickness of sputtering layer controlled at 300 nm. The electrode materials were proven not to have any obvious influence on the pyro-pyroelectric properties.

## Ferroelectric measurement

The polarization *versus* electric field (*P-E*) hysteresis loops were measured on a ferroelectric analyzer (Radiant Precision Premier II) employing the Sawyer-Tower circuit method. To avoid electric discharge at a high electric field, a single-crystal electrode was immersed in silicone oil to measure the *P-E* hysteresis loops.

## Pyroelectric and photo-pyroelectric measurements

The stable pyroelectric current was measured using a Keithley 2636B electrometer test system in the heating process with a constant heating rate. And pyroelectric coefficients were analyzed by an indirect method of charge integration. To apply a cyclic change in pyroelectric current on N-IBATFA, a laser with a range of wavelengths was adopted on a lateral structure device under zero bias. The current oscillations were achieved by switching the laser periodically on and off. A Keithley 2636B electrometer was used to measure the output current. The dynamic pyroelectric current was obtained at room temperature using the Chynoweth technique. THORLABS 405, 520, 637, 785, and 940 nm pigtailed laser diodes (LP405-MF300, LP520-MF100, LP637-SF70, LP785-SF100, LP940-SF30) were used for the visible light illumination. The homemade 266 nm laser (LDMQ-266-10) was used for the ultraviolet light illumination. The New Focus 1550 nm and homemade 1950 nm lasers (Velocity TLB-6730, LDM-1950-20) were used for the infrared light illumination. The incident light intensity was measured by the light power meter. Thermal imaging photos were taken with a HIKVISION H21pro camera.

## Data availability

All relevant data are presented via this publication and Supplementary Information. The X-ray crystallographic coordinates for structures reported in this study have been deposited at the Cambridge Crystallographic Data Centre (CCDC), under deposition numbers 2235608 and 2235609. These data can be obtained free of charge from The Cambridge Crystallographic Data Centre via www.ccdc.cam.ac.uk/getstructures. The data that support this study are available from the corresponding author upon request.

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

## Acknowledgements

This work was supported by the National Natural Science Foundation of China (22125110, 22205233, 21833010, 22193042, 21921001 and U21A2069), the National Key Research and Development Program of China (2019YFA0210402), the Key Research Program of Frontier Sciences of CAS (ZDBS-LY-SLH024), the Strategic Priority Research Program of CAS (XDB20010200), and Fujian Science & Technology Innovation Laboratory for Optoelectronic Information of China (2021ZR126). We are grateful to Prof. Xiaoying Huang for assistance in crystal structure analysis.

## Author contributions

The manuscript was written through contributions of all authors. X.Z. and Z.S. conceived the project. X.Z. designed and performed most of the experiments and analyzed the data. Y.L., L.H., L.T., W.G., Y.C., T.Y., and H.X. offered help in data analyses. W.W. offered help in some experiments. X.Z. prepared the manuscript. Z.S. revised the manuscript. J.L. and Z.S. supervised the project.

## Competing interests

The authors declare no competing interests.
