## [Peer Review File · Nature Communications]

A molecular pyroelectric enabling broadband photo-pyroelectric effect towards self-driven wide spectral photodetectionREVIEWER COMMENTS

Reviewer #1 (Remarks to the Author):

In this paper, the authors reported the first metal-free molecular pyroelectric compound N-IBATFA with UV-NIR spectral photo-pyroelectric effect. This compound possesses a large P_s , high pyroelectric coefficient and figures-of-merits. This discovery enriches the metal-free pyroelectric family, which broadens the application range of pyroelectric materials to a certain extent. In addition, the authors present a detailed study of the characteristics of N-IBATFA, and a logical description is made. Therefore, I recommend acceptance of this manuscript in Nature Communications after major revision.

1. In this work, the pyroelectric current is induced by the laser, which varies with a temperature change generated by alternatively switching light. However, the laser here can only act as the heat source. The authors claimed that this is a thermal process almost independent of the optical absorption of pyroelectric materials, thus allowing the wide spectral photoresponse beyond their energy bandgap. Additionally, this material seems do not show specific recognition of light at different wavelengths. As the wavelength and power density of the laser change simultaneously, it is possible to obtain the same current peak. Since there is no interaction between the thermal-induced pyroelectric effect and light irradiation in N-IBATFA materials, these properties are considered independent, and it is difficult to consider them as new functional materials.

2. For organic salts, the concept of metal-free is nonsense, and the authors should delete this description.

3. The authors compared the properties such as polarization, pyroelectric coefficient, and figures-of-merits of N-IBATFA and other typical pyroelectric materials.

a. Please explain why this material is better or worse than other pyroelectric materials.

b. Besides, the reported compound shows many advantages over the most typical pyroelectric material TGS. I think it would be attractive if the authors can make a comparison of the broadband photo-pyroelectric effect between these two excellent materials.

4. The authors should highlight the advantage of light irradiation over thermal control in the Introduction section.

5. The authors omitted the label of atom in Fig 2 and Fig. S6.

6. In the methods section, is "Pyro-pyroelectric measurement" a typo?

Reviewer #2 (Remarks to the Author):

This manuscript is interesting solely with respect to the high pyrocoefficient reported on the metal-free molecular pyroelectric system. However, I have major concerns about their claim about the broadband photo-pyroelectric effect and their proposed application as wide spectral photodetector. The authors should address the following queries before taking final decision about this manuscript.

1. I think this is the first time the authors have synthesized the N-IBATFA molecular pyroelectric crystals. If so, it should have been clearly brought out in the manuscript which is in my opinion not very clear in

the present version. If this is not the first time, the relevant literature should be quoted.

2. There are many metal-free molecular ferroelectric systems available in the literature. In this aspect, the reason for the selection of this material is missing in the introduction.

3. Those reported systems have provided the direct polarization versus electric field hysteresis loops, however, it is not given in this manuscript given the fact that the authors have grown single crystal of N-IBATFA with sufficient dimension. Is it only pyroelectric system and not ferroelectric?

4. The (BA)₂(EA)₂Pb₃I₁₀ (reference number 12 in supplementary section) exhibits better pyroelectric response near room temperature (315 K) than the value reported in this manuscript. Moreover, the maximum pyroelectric response of the present system is only at 360 K which restrict their room temperature application. Kindly comment on this.

5. The major concern is about the photopyroelectric response measured on light ON/OFF conditions over wide spectral region. It is clearly mentioned in the manuscript that the light source is used to generate the pyroelectric signal by heating the sample. As we know, the pyroresponse is related to dT/dt. In this perspective, I would expect the sample to display higher pyrocurrent response in IR region than the UV region. Looking at the bandgap of the system (4.46 eV) which is falling under UV region, the most of the UV light is likely to get absorbed in the process of generating electron-hole pair, due to the well-known photovoltaic effect, rather than used to heat up the system. Contrary to this, the Fig.4b shows the opposite trend. The authors should give clear explanation for their observed wavelength dependent photo-pyroelectric response.

6. The authors should provide the dT/dt data measured under light on-off condition which could through more light on the heating effect.

7. The purpose of figure 3c, where the thermal imaging photos of crystal surface under dark and light illuminated conditions are given, is not serving any purpose. The authors should provide the better image where the clear contrast between different temperature region could be seen.

8. In the calculation of photodetector parameters like R and D, the authors seem to use the photocurrent I_{ph} . Is it a photocurrent or pyrocurrent? Are they same? More clarity is needed.

Overall, the pyroelectric response part of the manuscript is interesting, however, the photo-pyroelectric response towards achieving wide band photodetector is not very convincing. Hence, I am not recommending this manuscript for publication in the present form.

Reviewer #3 (Remarks to the Author):

The review has been prepared according to the following six questions.

1. What are the noteworthy results?

In the following manuscript, Authors obtained and characterized a metal-free molecular pyroelectric N-isopropylbenzylammonium trifluoroacetate (N-IBATFA), of which broadband photo-pyroelectric effects allow for self-driven wide spectral photodetection. The results are sound and well-presented. The phase

situation (one reversible phase transition) was detected by DSC measurement, while the non-centrosymmetric character of LTP was confirmed by SHG analysis. Based on pyroelectric current analysis the P_s value was estimated. The most interesting is the paragraph in which the broadband photo-pyroelectric effect is studied. These results are prepared with great care and precision.

2. Will the work be of significance to the field and related fields? How does it compare to the established literature? If the work is not original, please provide relevant references.

I have no doubts that the presented results are significant in widely understood material chemistry or materials engineering.

3. Does the work support the conclusions and claims, or is additional evidence needed?

In my opinion, this paper supports the conclusions and claims; no additional evidence is needed.

4. Are there any flaws in the data analysis, interpretation and conclusions? Do these prohibit publication or require revision?

In my opinion, the publication should be published in its current form.

5. Is the methodology sound? Does the work meet the expected standards in your field?

Yes

6. Is there enough detail provided in the methods for the work to be reproduced?

Yes

Itemized response to the Reviewers' comments

Reviewer #1 (Remarks to the Author):

In this paper, the authors reported the first metal-free molecular pyroelectric compound *N*-IBATFA with UV-NIR spectral photo-pyroelectric effect. This compound possesses a large P_s , high pyroelectric coefficient and figures-of-merits. This discovery enriches the metal-free pyroelectric family, which broadens the application range of pyroelectric materials to a certain extent. In addition, the authors present a detailed study of the characteristics of *N*-IBATFA, and a logical description is made. Therefore, I recommend acceptance of this manuscript in *Nature Communications* after major revision.

Response: We deeply thank the reviewer for the time in reviewing the paper and professional feedback. The questions and suggestions are very important to improve our work. We have carefully revised our manuscript according to these constructive suggestions.

1. In this work, the pyroelectric current is induced by the laser, which varies with a temperature change generated by alternatively switching light. However, the laser here can only act as the heat source. The authors claimed that this is a thermal process almost independent of the optical absorption of pyroelectric materials, thus allowing the wide spectral photoresponse beyond their energy bandgap. Additionally, this material seems do not show specific recognition of light at different wavelengths. As the wavelength and power density of the laser change simultaneously, it is possible to obtain the same current peak. Since there is no interaction between the thermal-induced pyroelectric effect and light irradiation in *N*-IBATFA materials, these properties are considered independent, and it is difficult to consider them as new functional materials.

Response: Thanks for the reviewer's valuable comment. During our pyroelectric measurement, the generation of pyroelectric current closely involves the thermally-induced variation of electric polarization, thus allowing the wide spectral photoresponse beyond the optical absorption. However, it is still possible to qualitatively estimate the magnitude of current under different light illumination. We here performed a systematic investigation to clarify the potential relationship between the pyroelectric current peak (I_{pyro}) and power density or laser wavelength in numerical terms. Firstly, we measured the I_{pyro} with respect to the incident power density at the fixed wavelengths. The results indicate that an almost linear relationship exists between I_{pyro} and power density under laser irradiation at various wavelengths. Fig. R1 shows the almost linear function, using the examples of 405 nm and 520 nm wavelength lasers. Consequently, we might infer the power density of a specific laser wavelength based on the observed I_{pyro} . Furthermore, we investigated the relationship between I_{pyro} and laser wavelength at fixed power density and found that there is an exponential relationship

between I_{pyro} and wavelength under laser irradiation at the selected wavelengths. Fig. R2 illustrates this relationship for power densities of approximately 130 mW cm^{-2} and 160 mW cm^{-2} , respectively. Hence, we can deduce the wavelength of a particular laser by analyzing I_{pyro} when irradiated at a fixed power density. These findings reveal that *N*-IBATFA might be a potential functional material capable of discerning the power density at a specific incident light wavelength and the incident light wavelength at a specific power density.

Fig. R1. The photoexcited pyroelectric current peak as a function of power density under 405 nm (a) and 940 nm (b) laser irradiation.

Fig. R2. The photoexcited pyroelectric current peak as an exponential function of incident wavelength with a power density of around 130 mW cm^{-2} (a) and 160 mW cm^{-2} (b).

2. For organic salts, the concept of metal-free is nonsense, and the authors should delete this description.

Response: Thanks for the reviewer's comment. According to the suggestion, we have deleted the description "metal-free" in the revision. Compared with the metal-containing counterparts, the absence of metal should be an advantage of our work.

3. The authors compared the properties such as polarization, pyroelectric coefficient, and figures-of-merits of *N*-IBATFA and other typical pyroelectric materials.

a. Please explain why this material is better or worse than other pyroelectric materials.

Response: *N*-IBATFA exhibits comparable pyroelectric behaviors with some other counterparts, while it has unique characteristics for potential applications. Firstly, in comparison to inorganic oxide materials, *N*-IBATFA, as a simple organic binary salt, could be easily synthesized from the aqueous solution. The absence of metal affords possible advantages of non-toxicity, light weight, flexibility, biocompatibility, and ease of processing (*Science* **1983**, 220, 1115-1121; *J. Am. Chem. Soc.* **2019**, 141, 9349-9357; *J. Phys. Chem. Lett.* **2019**, 10, 6650-6655). Secondly, a large polarization (P_s) value serves as an indicator of strong polarity, which signifies the potential of *N*-IBATFA as a pyroelectric material (*Phys. Today* **2005**, 58, 31; *Adv. Mater.* **2012**, 24, 5357-5362). Thirdly, *N*-IBATFA demonstrates a notable pyroelectric coefficient (P_e), implying that a larger pyroelectric current (I_{pyro}) can be output at a certain rate of temperature change due to the positive relationship between P_e and I_{pyro} (*Ferroelectrics* **1976**, 10, 83-89; *Energy Environ. Sci.* **2014**, 7, 3836-3856). Lastly, since there is an inverse relationship between figures-of-merits (FOMs) and ϵ_r , the smaller dielectric constant (ϵ_r) of *N*-IBATFA allows to output a higher voltage. Voltage responsivity (F_v) is a crucial parameter for evaluating the efficiency of electric voltage output efficiency, and a high F_v value is vital for enhancing the sensitivity of pyroelectric devices (*Sci. Adv.* **2021**, 7, eabe3068; *The Innovation* **2022**, 3, 100204). Therefore, it is believed that *N*-IBATFA might be a potential pyroelectric material with unique merits, such as non-toxicity, light-weight and flexible nature, along with large polarization and lower dielectric constant.

b. Besides, the reported compound shows many advantages over the most typical pyroelectric material TGS. I think it would be attractive if the authors can make a comparison of the broadband photo-pyroelectric effect between these two excellent materials.

Response: We concur with the consensus that TGS is a very excellent material for pyroelectric detection and infrared sensing. Extensive research has been conducted on TGS and its doped compounds, particularly focusing on enhancing the Curie temperature through chemical modification. In our study, our main objective is to explore a new pyroelectric candidate of *N*-IBATFA, which has wide spectral photodetection capabilities. Under the same experimental conditions, we conducted a comparative analysis of the photo-pyroelectric responses between TGS and *N*-IBATFA crystals, as depicted in Fig. R4. The results reveal that *N*-IBATFA exhibits photo-pyroelectric response comparable to that of TGS. For instance, upon irradiation with 520 nm laser at a power density of 167 mW cm⁻², TGS demonstrates a sharp transient current spike (I_{pyro}) of ~0.01 nA, while *N*-IBATFA exhibits an I_{pyro} of ~0.012 nA under irradiation with a power density of 165 mW cm⁻².

Fig. R3. (a) Colorless large-size crystal of TGS. (b) Simulated and experimental PXRD patterns of TGS at room temperature. (c) Temperature dependence of pyroelectric current. (d) Variable-temperature polarization (P_s) determined by integrating pyroelectric current over time.

Fig. R4. Variations of current intensity as a function of power density at 405 nm and 520 nm for TGS (a) and *N*-IBATFA (b).

As the reviewer mentioned, the photo-pyroelectric response of TGS has been added in the revised manuscript and supplementary information.

Paragraph 3 page 4:

Moreover, to further verify the photo-pyroelectric behaviors, we conducted a comparative analysis of the light-induced pyroelectric responses between TGS and *N*-IBATFA. The qualitative results indicate that *N*-IBATFA exhibits photo-pyroelectric response comparable to that of TGS (Fig. S14).

Fig. S14 page S9:

Fig. S14. Variations of current intensity as a function of power density at 405 nm and 520 nm for TGS crystal.

4. The authors should highlight the advantage of light irradiation over thermal control in the Introduction section.

Response: Thanks for the reviewer's valuable comments. We have added "Furthermore, the photo-pyroelectric detector exhibits a substantially quicker response in comparison with the thermal-based detector, which might attribute to the direct correlation between photo-pyroelectric signal and temperature change, and its physical process does not necessitate the establishment of thermal balance."³³ in the revision (paragraph 3 page 1).

The related reference is presented as follows:

33. Sassi, U. et al. Graphene-based mid-infrared room-temperature pyroelectric bolometers with ultrahigh temperature coefficient of resistance. *Nat. Commun.* **8**, 14311 (2017).

5. The authors omitted the label of atom in Fig. 2 and Fig. S6.

Response: Many thanks for the reviewer's comment. We have added the labels of atoms in Fig. 4b and Fig. S6 as follows:

Fig. 2 | Crystal structures of *N*-IBATFA. a The reversible phase transition between LTP (left) and HTP structure (right) (H-bonds and glide planes are represented by magenta dotted lines and orange planes, respectively). b Hydrogen bonds of *N*-IBATFA at HTP. c Symmetry breaking with an Aizu notation of $2/mF2$.

Fig. S6. Hydrogen bonds of *N*-IBATFA at LTP.

6. In the methods section, is “Pyro-pyroelectric measurement” a typo?

Response: Thank you for reminding. We have revised “Pyro-pyroelectric measurement” to “Photo-pyroelectric measurement” in the methods section.

===== The end of reply to Reviewer 1 =====

Reviewer #2 (Remarks to the Author):

This manuscript is interesting solely with respect to the high pyrocoefficient reported on the metal-free molecular pyroelectric system. However, I have major concerns about their claim about the broadband photo-pyroelectric effect and their proposed application as wide spectral photodetector. The authors should address the following queries before taking final decision about this manuscript.

Response: We deeply thank the reviewer for the time in reviewing our paper and professional feedback. The comments and suggestions are very important to improve our work. We have carefully revised our manuscript according to these constructive suggestions.

1. I think this is the first time the authors have synthesized the *N*-IBATFA molecular pyroelectric crystals. If so, it should have been clearly brought out in the manuscript which is in my opinion not very clear in the present version. If this is not the first time, the relevant literature should be quoted.

Response: Thank a lot for the reviewer's reminding. Before this study, we have checked the known crystal structures containing the *N*-isopropylbenzylaminium cation, especially in The Cambridge Crystallographic Data Centre (CCDC). To the best of our knowledge, this study should represent the first synthesis of the *N*-IBATFA molecular pyroelectric crystal. Most interestingly, *N*-IBATFA enables the broadband photo-pyroelectric effect, which sheds light on the self-driven wide spectral photodetection. As a new compound, we have also applied CCDC numbers 2235608 and 2235609 for the crystal structures of *N*-IBATFA. According to the reviewer's suggestion, we have described this point clearly in the revision. For instance:

The modified content in abstract:

Here, we present a new molecular pyroelectric, *N*-isopropylbenzylaminium trifluoroacetate (*N*-IBATFA), of which the broadband photo-pyroelectric effects allow for self-driven wide spectral photodetection.

The content modified on paragraph 1 page 2:

For the first time, we here present a new molecular pyroelectric, *N*-isopropylbenzylaminium trifluoroacetate (*N*-IBATFA), of which the photo-pyroelectric effect drives broadband spectral photoresponse from 266 to 1950 nm.

2. There are many metal-free molecular ferroelectric systems available in the literature. In this aspect, the reason for the selection of this material is missing in the introduction.

Response: Thanks for the reviewer's valuable comments. It is known that ferroelectric materials are the most promising pyroelectric candidates, for which the generation of electric polarization in the structures corresponds to paraelectric-ferroelectric phase transition. From a structural viewpoint, the dynamic motion of isopropyl moiety in *N*-isopropylbenzylaminium cation and the order-disordering

of CF_3COO^- anion might facilitate the phase transition. Besides, the introduction of fluorine has been proven as an effective strategy to improve phase transition in molecular ferroelectrics. That is why we choose this new compound as a pyroelectric candidate. We have made the related revision in the manuscript.

Paragraph 3 page 1:

In this study, we initially select a suitable polar component of *N*-isopropylbenzylamine (*N*-IBA), which is conducive to inducing the occurrence of ferroelectric phase transition and symmetry breaking.³⁴ The order-disordering feature of CF_3COO^- cation also facilitates the phase transition. Meanwhile, the F substitution has been proven as an effective strategy to achieve the relatively high Curie temperature (T_c).³⁵⁻³⁶

The related references are presented as follows:

34. Sun, Z. et al. Unusual long-range ordering incommensurate structural modulations in an organic molecular ferroelectric. *J. Am. Chem. Soc.* **139**, 15900-15906 (2017).

35. Ai, Y, et al. Fluorine substitution induced high T_c of enantiomeric perovskite ferroelectrics: (*R*)- and (*S*)-3-(fluoropyrrolidinium) MnCl_3 . *J. Am. Chem. Soc.* **141**, 4474-4479 (2019).

36. Tang, Y. Y, et al. Record enhancement of phase transition temperature realized by H/F substitution. *Adv. Mater.* **32**, 2003530 (2020).

3. Those reported systems have provided the direct polarization *versus* electric field hysteresis loops, however, it is not given in this manuscript given the fact that the authors have grown single crystal of *N*-IBATFA with sufficient dimension. Is it only pyroelectric system and not ferroelectric?

Response: Thanks a lot for the reviewer's comments. For ferroelectrics, the electric polarization *versus* electric field (*P-E*) hysteresis loop behaves as a direct sign of ferroelectricity, namely, the polarization switching under an external field. As the reviewer suggested, we have also performed the *P-E* measurement along the polar axis of crystal sample, and the details are provided in the revised manuscript and supplementary information.

Paragraph 3 page 2:

The electric polarization *versus* electric field (*P-E*) hysteresis loop of *N*-IBATFA crystal was measured along its polar axis at LTP. However, the electric switching of polarization could not be achieved under a high electric field up to ~120 kV/cm (Fig. S8). Further studies on the domain structures are still needed in the future. Consequently, the current evidence is insufficient to conclusively confirm its ferroelectricity, while the ferroelectric phase transition indicates that *N*-IBATFA is expected to be a potential metal-free molecular pyroelectric material.

In the method section:

Ferroelectric measurement. The polarization *versus* electric field (*P-E*) hysteresis loops were measured on a ferroelectric analyzer (Radiant Precision Premier II) employing the Sawyer-Tower circuit method. To avoid electric discharge at a high electric field, a single crystal electrode was immersed in silicone oil to measure the *P-E* hysteresis loops.

Fig. S8 page S5:

Fig. S8. The *P-E* hysteresis loop measured along the polar axis of *N*-IBATFA crystal at LTP.

The structural phase transition of *N*-IBATFA, as indicated by DSC, dielectric measurement, SHG symmetry breaking and crystal structure analyses, obeys the ferroelectric symmetry breaking with an Aizu notation of $2/mF2$ (*J. Phys. Soc. Jpn.* **1969**, *27*, 387-389; *Chem. Soc. Rev.* **2016**, *45*, 3811-3827). However, the *P-E* hysteresis loop obtained from the single crystal along the polar axis direction was the linear trace (at ~ 345 K). Even with an external electric field approaching the limit of 10^4 V (~ 120 kV/cm), the electric domains still could not undergo flipping. This is potentially due to the ultrahigh electric coercivity field for the *N*-IBATFA crystal. At present, there is insufficient evidence to conclusively confirm its ferroelectricity, while we classify *N*-IBATFA as a pyroelectric material in the revision.

4. The $(\text{BA})_2(\text{EA})_2\text{Pb}_3\text{I}_{10}$ (reference number 12 in supplementary section) exhibits better pyroelectric response near room temperature (315 K) than the value reported in this manuscript. Moreover, the maximum pyroelectric response of the present system is only at 360 K which restrict their room temperature application. Kindly comment on this.

Response: Great thanks for the reviewer's kind comments. Compound $(\text{BA})_2(\text{EA})_2\text{Pb}_3\text{I}_{10}$ (reference 12 in Supplementary Section) does exhibit very intriguing pyroelectric responses near the room temperature (~ 315 K). As far as we know, this enhanced performance could be attributed to its unique

dielectric bistability, resulting from the improper mechanism of ferroelectric order, which leads to a temperature-independent variation of the dielectric constant in proximity to the Curie temperature (*Adv. Mater.* **2014**, *26*, 4515-4520; *Adv. Mater.* **2015**, *27*, 4795-4801). In contrast, the *N*-IBATFA crystal exhibits an average room-temperature pyroelectricity but displays excellent pyroelectric properties around 360 K. Furthermore, the emerging demand for environmentally friendly and sustainable utilization has motivated the exploration of non-toxic, cost-effective molecular pyroelectric alternatives or supplements to inorganic oxide counterparts (*Science* **1983**, *220*, 1115-1121; *J. Am. Chem. Soc.* **2019**, *141*, 9349-9357). Under this circumstance, we target to develop the metal-free, light-weight and environmentally friendly molecular pyroelectrics, as well as photo-pyroelectric materials, to provide a reference for flexible and biocompatible devices.

The corresponding content added on paragraph 1 page 4:

Overall, the superior figure-of-merits, including F_V and F_D , make *N*-IBATFA a potential material for pyroelectric application as the molecular alternative, and its non-toxic, light-weight and environmentally friendly advantages render it a valuable reference for the development of flexible and biocompatible devices.

5. The major concern is about the photopyroelectric response measured on light ON/OFF conditions over wide spectral region. It is clearly mentioned in the manuscript that the light source is used to generate the pyroelectric signal by heating the sample. As we know, the pyroresponse is related to dT/dt . In this perspective, I would expect the sample to display higher pyrocurrent response in IR region than the UV region. Looking at the bandgap of the system (4.46 eV) which is falling under UV region, the most of the UV light is likely to get absorbed in the process of generating electron-hole pair, due to the well-known photovoltaic effect, rather than used to heat up the system. Contrary to this, the Fig.4b shows the opposite trend. The authors should give clear explanation for their observed wavelength dependent photo-pyroelectric response.

Response: Thank for the reviewer's professional comments. We agree with the reviewer's statement that due to the photovoltaic effect, most of the UV light may be absorbed in the process of generating electron-hole pairs. The observed strong current signal in the UV light region is probably caused by the pyroelectric-photovoltaic coupled effects (*Adv. Mater.* **2017**, *29*, 1703694). Upon the UV-light illumination, a fast temperature increase occurs within pyroelectric *N*-IBATFA for light adsorption. The resulting photo-pyroelectric charge can effectively influence the charge transport across the interface and modulate the photovoltaic process of charge carriers (*Adv. Mater.* **2015**, *27*, 7963-7969). However, we do not observe a steady photovoltaic current plateau. It is suspected that this may be related to the device planar structure (*Adv. Mater.* **2017**, *29*, 1703694), or potentially due to the transient photovoltaic effect (*JPN J. Appl. Phys.* **2017**, *32*, 481; *Nat. Commun.* **2017**, *8*, 281).

Moreover, *N*-IBATFA exhibits a weaker pyroelectric response in IR region compared to UV region, which may be associated with the size of laser facula (0.38 cm² for IR and 0.13 cm² for UV). When the device is irradiated with a laser of same power density, the temperature increasing of device is inversely related to the size of laser facula. Further studies are still needed in the future.

The corresponding content added on paragraph 3 page 4:

The highest pyroelectric current response observed at 266 nm is possibly due to the coupling with photovoltaic effect, since the absorbed UV light might excite the generation of electron-hole pairs.⁴⁷⁻

48

The related reference is presented as follows:

47. Ma, N. et al. Photovoltaic–pyroelectric coupled effect induced electricity for self-powered photodetector system. *Adv. Mater.* **29**, 1703694 (2017).

48. Xun, H. et al. Piezo-phototronic enhanced UV sensing based on a nanowire photodetector array. *Adv. Mater.* **27**, 7963-7969 (2015).

6. The authors should provide the dT/dt data measured under light on-off condition which could through more light on the heating effect.

Response: We greatly appreciate the reviewer’s valuable comments. According to the suggestion, we have provided the dT/dt data measured under the light on-off conditions in the revised manuscript and supplementary information.

Paragraph 2 page 4:

The cyclic changes in temperature and the corresponding differential temperature dT/dt are shown in Fig. S9.

Fig. S9 page S6:

Fig. S9. The photoexcited pyroelectric current response, cyclic change in temperature, and the corresponding differential curve of *N*-IBATFA under a periodically switched 405 nm laser with power density of 160 mW cm⁻².

7. The purpose of figure 3c, where the thermal imaging photos of crystal surface under dark and light illuminated conditions are given, is not serving any purpose. The authors should provide the better image where the clear contrast between different temperature region could be seen.

Response: Thanks for the reviewer's valuable comments. According to your useful suggestion, the revised Figure 3c has been labeled with clearer thermal imaging photos. The sample temperature shows a variation from 0~4.5 K over a period of 10 s, under the 405 nm laser illumination of 160 mW cm⁻². The results are also coincident with the added Fig. S9.

The modified in Figure 3c:

Fig. 3c. Thermal imaging photos of crystal surface in the dark and irradiated by 405 nm laser.

8. In the calculation of photodetector parameters like R and D^* , the authors seem to use the photocurrent I_{ph} . Is it a photocurrent or pyrocurrent? Are they same? More clarity is needed.

Response: Thanks for the reviewer's careful reading of our manuscript. In calculating the photodetector parameters such as R and D^* , we used the peak value of photo-pyroelectric currents, which has been modified in the revised manuscript.

Overall, the pyroelectric response part of the manuscript is interesting, however, the photo-pyroelectric response towards achieving wide band photodetector is not very convincing. Hence, I am not recommending this manuscript for publication in the present form.

Response: We sincerely appreciate the valuable comments provided by the reviewer, which are quite helpful to improve our work. We have carefully revised our manuscript according to these constructive suggestions, and hope the revision would satisfy the publication standard.

=====**The end of reply to Reviewer 2**=====

Reviewer #3 (Remarks to the Author):

The review has been prepared according to the following six questions.

1. What are the noteworthy results?

In the following manuscript, Authors obtained and characterized a metal-free molecular pyroelectric N-isopropylbenzylaminium trifluoroacetate (*N*-IBATFA), of which broadband photo-pyroelectric effects allow for self-driven wide spectral photodetection. The results are sound and well-presented. The phase situation (one reversible phase transition) was detected by DSC measurement, while the non-centrosymmetric character of LTP was confirmed by SHG analysis. Based on pyroelectric current analysis the P_s value was estimated. The most interesting is the paragraph in which the broadband photo-pyroelectric effect is studied. These results are prepared with great care and precision.

2. Will the work be of significance to the field and related fields? How does it compare to the established literature? If the work is not original, please provide relevant references.

I have no doubts that the presented results are significant in widely understood material chemistry or materials engineering.

3. Does the work support the conclusions and claims, or is additional evidence needed?

In my opinion, this paper supports the conclusions and claims; no additional evidence is needed.

4. Are there any flaws in the data analysis, interpretation and conclusions? Do these prohibit publication or require revision?

In my opinion, the publication should be published in its current form.

5. Is the methodology sound? Does the work meet the expected standards in your field?

Yes.

6. Is there enough detail provided in the methods for the work to be reproduced?

Yes.

Response: Great thanks for the Reviewer's appraisal on this work. We have made detailed responses to all the reviewers' comments and suggestions.

===== **The end of reply to Reviewer 3** =====

REVIEWERS' COMMENTS

Reviewer #1 (Remarks to the Author):

I really appreciate the author's thorough response to my points. The correlation between the magnitude of the pyroelectric current of N-IBATFA and light illumination or power density is much clearer. The unique characteristics and advantages of N-IBATFA are also well demonstrated. Additionally, the analysis of the data presented is much deeper and provides better insight into these materials. There have been substantial improvements compared to the previous version. However, there were a couple of points I would like the authors to clarify before publishing the paper.

1. The authors measured the I_{pyro} with respect to the incident power density at the fixed wavelengths to clarify the potential relationship between the I_{pyro} and power density or laser wavelength in numerical terms. As shown in Fig. R1, there are linear relationships with different slopes that exist between I_{pyro} and power density under 405 nm and 940 nm laser irradiation. This indicates that there will be a point of intersection between the two simulated lines, which means that the wavelength of a particular laser cannot be deduced by analyzing I_{pyro} at the point of intersection when irradiated at a fixed power density. The authors should clearly emphasize this point and how to avoid that.
2. The authors mentioned that N-IBATFA is not ferroelectric, but they stated that the phase transition of the material is a ferroelectric phase transition on Page 2, Para 3, and these are contradictory. The authors should correct this statement.

Reviewer #2 (Remarks to the Author):

The authors addressed all the queries/concerns raised on their manuscript. Detailed experiments are carried out to clear the concerns. I agree with the authors that the P-E loops is difficult to achieve in their system due to the high field involved in their organic system. However, they can try PFM phase and amplitude curve to show the switchable nature of the polarization. This is just a suggestion. The revised manuscript can be accepted for publication.

Response to reviewers

Reviewer #1 (Remarks to the Author):

I really appreciate the author's thorough response to my points. The correlation between the magnitude of the pyroelectric current of N-IBATFA and light illumination or power density is much clearer. The unique characteristics and advantages of N-IBATFA are also well demonstrated. Additionally, the analysis of the data presented is much deeper and provides better insight into these materials. There have been substantial improvements compared to the previous version. However, there were a couple of points I would like the authors to clarify before publishing the paper.

Response: We thank the Reviewer for their positive assessment of our work.

1. The authors measured the I_{pyro} with respect to the incident power density at the fixed wavelengths to clarify the potential relationship between the I_{pyro} and power density or laser wavelength in numerical terms. As shown in Fig. R1, there are linear relationships with different slopes that exist between I_{pyro} and power density under 405 nm and 940 nm laser irradiation. This indicates that there will be a point of intersection between the two simulated lines, which means that the wavelength of a particular laser cannot be deduced by analyzing I_{pyro} at the point of intersection when irradiated at a fixed power density. The authors should clearly emphasize this point and how to avoid that.

Response: Thanks for the reviewer's valuable comment. In practical applications, the wavelength of the incident laser illumination is generally fixed. It can be seen from our experimental results that for a specific laser wavelength irradiated at a fixed power density, there is a clear relationship equation with the obtained I_{pyro} . If for other incident wavelengths, we could also perform the calibration between the measured I_{pyro} and power density. Therefore, the fixed wavelength of incident laser can be determined from the observed I_{pyro} value.

2. The authors mentioned that N-IBATFA is not ferroelectric, but they stated that the phase transition of the material is a ferroelectric phase transition on Page 2, Para 3, and these are contradictory. The authors should correct this statement.

Response: Thanks a lot for the reviewer's comments. From the perspective of symmetry breaking, the structural phase transition of N-IBATFA belongs to one of the 88 species of paraelectric-to-ferroelectric phase transition with an Aizu notation of $2/mF2$ (*J. Phys. Soc. Jpn.* **1969**, *27*, 387-389; *Chem. Soc. Rev.* **2016**, *45*, 3811-3827). This symmetry breaking can be further verified by DSC, dielectric measurement, variable temperature SHG and crystal structures of N-IBATFA. However, ferroelectrics are not distinguished solely by symmetry considerations, but also verified by experimental features of electric switching of spontaneous polarization such as ferroelectric

hysteresis loops and electric domain motions. At present, we could not obtain the characteristic P - E hysteresis loops to confirm its ferroelectric properties. Moreover, as an effective method to verify ferroelectricity, our PFM measurement on N -IBATFA crystal also shows no motions of electric domain, revealing the absence of its solid ferroelectricity. Thus, we can not confirm that N -IBATFA is a verified ferroelectric, while the direct evidence of polarization switching is still lacking, and more experimental characterization is needed in the future.

Reviewer #2 (Remarks to the Author):

The authors addressed all the queries/concerns raised on their manuscript. Detailed experiments are carried out to clear the concerns. I agree with the authors that the P - E loops is difficult to achieve in their system due to the high field involved in their organic system. However, they can try PFM phase and amplitude curve to show the switchable nature of the polarization. This is just a suggestion. The revised manuscript can be accepted for publication.

Response: We thank the Reviewer for their positive assessment of our work. The previous results have shown that the structural phase transition of N -IBATFA belongs to one of the 88 species of paraelectric-to-ferroelectric phase transition with an Aizu notation of $2/mF2$ from the perspective of symmetry breaking (*J. Phys. Soc. Jpn.* **1969**, 27, 387-389; *Chem. Soc. Rev.* **2016**, 45, 3811-3827), which can be further verified by variable temperature SHG result. We agree with the statement that PFM is an effective method to verify ferroelectricity. However, our PFM measurements on N -IBATFA crystal show no motions of electric domain, coinciding with the observed linear P - E hysteresis loop, which reveals the current absence of its solid ferroelectricity. More experimental characterization is needed in the future.